# Bifurcation Analysis of a Duopoly Game with R&D Spillover, Price Competition and Time Delays

**B. A. Pansera [1], L. Guerrini [2] , M. Ferrara [1,3,\*] and T. Ciano [1,4,]**

[1] Department of Law, Economics and Human Sciences & Decision_lab, "Mediterranea" University of Reggio Calabria, via dell'Università, 25, 89124 Reggio Calabria, Italy; bruno.pansera@unirc.it (B.A.P.); up966762@myport.ac.uk (T.C.)

[2] Department of Management, Polytechnic University of Marche, Piazzale Martelli 8, 60121 Ancona, Italy; luca.guerrini@univpm.it

[3] Research Affiliate ICRIOS – The Invernizzi Centre for Research in Innovation, Organization, Strategy and Entrepreneurship Bocconi University – Department of Management and Technology, Via Sarfatti, 25, 20100 Milano, Italy

[4] Faculty of Business and Law, University of Portsmouth, Richmond Building, Portland Street, Portsmouth PO1 3DE, UK

\* Correspondence: massimiliano.ferrara@unirc.it

**Abstract:** The aim of this study is to analyse a discrete-time two-stage game with R&D competition by considering a continuous-time set-up with fixed delays. The model is represented in the form of delay differential equations. The stability of all the equilibrium points is studied. It is found that the model exhibits very complex dynamical behaviours, and its Nash equilibrium is destabilised via Hopf bifurcations.

**Keywords:** duopoly; Hopf bifurcation; R&D; time delay

## 1. Introduction

Research and development (R&D) is one of the main strengths of firms growth. Firms need to pursue R&D as an effective way to reduce production costs and improve quality of products, so as to increase the competitiveness of firms in the market [1]. R&D behaviour is eventually followed by R&D spillover. R&D spillovers are likely because of the exchange of information on R&D between firms and the distribution of human resources. Over the last few years, the topic of competitiveness and collaboration throughout R&D spending has drawn growing interest from entrepreneurs and economists. The AJ model proposed by d'Aspremont and Jacquemin [2] and the KMZ model proposed by Kamien et al. [3] are two representative models for simulating the spillover effect of R&D. Such two models are two-stage game models and, respectively, addressed the spillover of R&D and the spillover of R&D production. Nowadays, the two-stage game has attracted the attention of many academics. Bischi and Lamantia [4,5] suggested a two-stage system to represent firms R&D networks in the marketplace. Matsumura et al. [6] proposed a two-stage Cournot model where companies select R&D spending at the first step and choose production amounts at the second stage. Shibata [7] analysed spillovers of R&D spending across different market structures. In particular, he expanded the work of Matsummura et al. [6] to integrate R&D investment spillovers. The implementation of chaos theory in structural dynamic economics developed by Day [8] presented a theoretical basis for the analysis of a complex model. The synthesis of dynamic theory and oligopoly theory has become a primary tool for economists and mathematicians to research economic phenomena. In recent years, it has attracted the attention of a growing number of researchers to investigate the evolution of the economic system and describe the complex economic phenomenon using chaos theory. Gangopadhyay [9] developed a

complex model of enterprise merger, exploring bifurcation activity and multiple attractors coexistence in the designed environment. Li and Ma [10] considered a small rational dual-channel game and simulate their model's complex dynamic behaviour in their research. Many researchers have explored the complex dynamical behaviours of this type of models from different aspects, such as differentiated goods [11–15], bounded rationality [16], heterogeneous firms [7,17–19], delayed decisions [20–25] and other factors [26–28].

In this paper, we reconsider the discrete duopoly game model of R&D competition between two high-tech enterprises as introduced by Zhou and Wang [29], where the combination of game theory and nonlinear dynamics theory is applied to a monopoly market with R&D spillover. Their model happens to be described by

$$
\begin{aligned}
x_1(t+1) &= x_1(t) + \alpha_1 x_1(t) \left\{ \frac{4Bb}{9} \left[ x_1(t) + x_2(t) \right] + \frac{4A}{9} - \gamma x_1(t) \right\}, \\
x_2(t+1) &= x_2(t) + \alpha_2 x_2(t) \left\{ \frac{4Bb}{9} \left[ x_1(t) + x_2(t) \right] + \frac{4A}{9} - \gamma x_2(t) \right\},
\end{aligned}
\tag{1}
$$

where

$$
A = (a - 2bc)(\beta + 1), \quad B = (\beta + 1)^2.
$$

There are two firms, labelled by $m$ ($m = 1, 2$), in a market, which conduct R&D and produce complementary goods. Here, $x_m$ is the R&D effort of firm $m$. The parameters $a > 0$, $b > 0$ and $c > 0$ represent the market size, the price sensitivity of consumers and the unit cost of produced goods without R&D efforts, respectively. $\beta \in (0, 1)$ is related to the R&D spillover, whereas $\gamma > 0$ is the cost parameter of firm's technological innovation, which indicates the efficiency of using or producing the unique technology or knowledge resources for an enterprise. The smaller the parameter, $\gamma$, the stronger the innovation ability of firm $m$. Finally, $\alpha_m > 0$ is the speed of adjustment for firm $m$. A symmetry of parameters $\alpha_1$ and $\alpha_2$ exists in this system. Assuming continuous time scales and replacing $x_m(t+1) - x_m(t)$ ($m = 1, 2$) in (1) with $\dot{x}_m(t) = dx_m(t)/dt$, system (1) can be transformed into a continuous-time model, which may be further extended to a dynamic environment characterised by differential equations with two fixed delays. Within this framework, we show how the introduction of delays may cause chaotic dynamics that cannot be observed when time delays are absent, therefore providing a starting point for building on more sophisticated models with R&D.

The structure of this article is organised as follows. In Section 2, the continuous two-stage Cournot model with R&D spillover is established. In Section 3, the corresponding model with time delays is considered. The stability of its equilibrium points is discussed in case of one or two delays, and the occurrence of Hopf bifurcations is shown. Section 4 outlines the conclusions.

## 2. Continuous-Time Dynamics Model

After a simple algebraic manipulation, system (1) can be rewritten as

$$
\begin{aligned}
\dot{x}_1(t) &= \alpha_1 x_1(t) \left\{ \frac{4Bb}{9} \left[ x_1(t) + x_2(t) \right] + \frac{4A}{9} - \gamma x_1(t) \right\}, \\
\dot{x}_2(t) &= \alpha_2 x_2(t) \left\{ \frac{4Bb}{9} \left[ x_1(t) + x_2(t) \right] + \frac{4A}{9} - \gamma x_2(t) \right\}.
\end{aligned}
\tag{2}
$$

Noticing that the steady states of (2) are the same as the ones of system (1), from work in [29] we know that there exist three equilibrium points

$$
E_0 = (0, 0), \quad E_1 = \left( 0, \frac{4A}{9\gamma - 4Bb} \right), \quad E_2 = \left( \frac{4A}{9\gamma - 4Bb}, 0 \right)
$$

and a Nash–Cournot equilibrium point

$$E_3 = \left( \frac{4A}{9\gamma - 8Bb}, \frac{4A}{9\gamma - 8Bb} \right).$$

To guarantee the economic meaningfulness of these equilibria, we assume the conditions

$$a > 2bc, \quad \gamma > 8Bb.$$

Let $E_* = (x_1^*, x_2^*)$ denote a steady state of (2). The local stability of $E_*$ is governed by the roots of the corresponding characteristic equation for (2). By linearising (2) at $E_*$, we obtain the Jacobian matrix

$$J_{E_*} = \begin{bmatrix} \left( \dfrac{4Bb}{9} - \gamma \right) \alpha_1 x_1^* & \dfrac{4Bb\alpha_1 x_1^*}{9} \\[3mm] \dfrac{4Bb\alpha_2 x_2^*}{9} & \left( \dfrac{4Bb}{9} - \gamma \right) \alpha_2 x_2^* \end{bmatrix}.$$

It is well-known that stable solutions occur if and only if both eigenvalues of $J_{E_*}$ have negative real part, and this happens exactly when the trace of $J_{E_*}$ is negative, i.e.,

$$tr(J_{E_*}) = \left( \frac{4Bb}{9} - \gamma \right) (\alpha_1 x_1^* + \alpha_2 x_2^*) < 0, \tag{3}$$

and the determinant of $J_{E_*}$ is positive, i.e.,

$$\det(J_{E_*}) = \left( -\frac{8Bb}{9} + \gamma \right) \gamma \alpha_1 \alpha_2 x_1^* x_2^* > 0. \tag{4}$$

**Lemma 1.** *$E_0, E_1$ or $E_2$ are unstable equilibrium points, whereas $E_3$ is a stable equilibrium point.*

**Proof.** As the equilibrium point $E_1$ is symmetric with $E_2$ in the rectangular coordinate system, their stability analysis is very similar. When $E_* = E_1$ or $E_* = E_2$, one has $tr(J_{E_*}) = \det(J_{E_*}) = 0$. On the other hand, when $E_* = E_3$, we see that $tr(J_{E_*}) < 0$ and $\det(J_{E_*}) > 0$. The conclusion is then straightforward. □

## 3. Delay Dynamics Model

We now transform the discrete-time model (1) into a continuous-time model with delays by following the approach of [30], and derive the following two-dimensional system with distinct time delays $\tau_1 \geq 0, \tau_2 \geq 0$,

$$
\begin{aligned}
\dot{x}_1(t) &= \alpha_1 x_1(t - \tau_1) \left\{ \frac{4Bb}{9} [x_1(t - \tau_1) + x_2(t - \tau_1)] + \frac{4A}{9} - \gamma x_1(t - \tau_1) \right\}, \\
\dot{x}_2(t) &= \alpha_2 x_2(t - \tau_2) \left\{ \frac{4Bb}{9} [x_1(t - \tau_2) + x_2(t - \tau_2)] + \frac{4A}{9} - \gamma x_2(t - \tau_2) \right\}.
\end{aligned}
\tag{5}
$$

It is clear that the equilibrium points $E_* = (x_1^*, x_2^*)$ of (5) coincide with those of system (1).

*3.1. Existence of Equilibria and Local Bifurcations with Homogeneous Time Delays*

By setting $\tau_1 = \tau_2 = \tau \geq 0$, system (5) becomes

$$
\begin{aligned}
\dot{x}_1(t) &= \alpha_1 x_1(t-\tau) \left\{ \frac{4Bb}{9} [x_1(t-\tau) + x_2(t-\tau)] + \frac{4A}{9} - \gamma x_1(t-\tau) \right\}, \\
\dot{x}_2(t) &= \alpha_2 x_2(t-\tau) \left\{ \frac{4Bb}{9} [x_1(t-\tau) + x_2(t-\tau)] + \frac{4A}{9} - \gamma x_2(t-\tau) \right\}.
\end{aligned}
\tag{6}
$$

To examine the stability of $E_*$, we consider the characteristic equation of the linearisation of (6) at $E_* = (x_1^*, x_2^*)$ and get

$$
\begin{vmatrix}
-\lambda + \left[ \left( \frac{4Bb}{9} - \gamma \right) \alpha_1 x_1^* \right] e^{-\lambda\tau} & \left( \frac{4Bb\alpha_1 x_1^*}{9} \right) e^{-\lambda\tau} \\
\left( \frac{4Bb\alpha_2 x_2^*}{9} \right) e^{-\lambda\tau} & -\lambda + \left[ \left( \frac{4Bb}{9} - \gamma \right) \alpha_2 x_2^* \right] e^{-\lambda\tau}
\end{vmatrix} = 0,
$$

which writes as

$$
\lambda^2 - tr(J_{E_*})\lambda e^{-\lambda\tau} + \det(J_{E_*})e^{-2\lambda\tau} = 0.
\tag{7}
$$

If all the roots of (7) have negative real parts, then the equilibrium $E_*$ of (6) is locally asymptotically stable, and it is unstable if (7) has at least one root with positive real part. In case $\tau = 0$, assume that $E_*$ is stable. Let $\tau > 0$. For computational purpose, we multiply both sides of (7) by $e^{\lambda\tau}$ and get

$$
\lambda^2 e^{\lambda\tau} - tr(J_{E_*})\lambda + \det(J_{E_*})e^{-\lambda\tau} = 0.
\tag{8}
$$

We use this equation to yield purely imaginary roots $i\omega$ to the characteristic Equation (7). Substituting $\lambda = i\omega$ ($\omega > 0$) into (8), we derive

$$
-\omega^2 e^{i\omega\tau} - tr(J_{E_*})i\omega + \det(J_{E_*})e^{-i\omega\tau} = 0.
\tag{9}
$$

Using $e^{i\omega\tau} = \cos\omega\tau + i\sin\omega\tau$, $e^{-i\omega\tau} = \cos\omega\tau - i\sin\omega\tau$, we separate the real and imaginary parts of (9) and find that $\omega$ satisfies

$$
\begin{cases}
\left[ -\omega^2 + \det(J_{E_*}) \right] \cos\omega\tau &= 0, \\
\left[ \omega^2 + \det(J_{E_*}) \right] \sin\omega\tau &= -tr(J_{E_*})\omega.
\end{cases}
\tag{10}
$$

**Proposition 1.** *System* (10) *has no positive solution if* $E_* = E_0$, *it has a positive root given by*

$$
\omega_0 = -tr(J_{E_*}), \qquad \tau = \tau_0^j = \frac{1}{\omega_0} \left( \frac{\pi}{2} + 2j\pi \right),
$$

*where* $j = 0, 1, 2, \ldots$, *if* $E_* = E_1$ *or* $E_* = E_2$, *or by*

$$
\omega_1 = \sqrt{\det(J_{E_*})}, \qquad \tau = \tau_1^j = \frac{1}{\omega_1} \arcsin \left[ -\frac{tr(J_{E_*})\omega_1}{\omega_1^2 + \det(J_{E_*})} \right] + 2j\pi
\tag{11}
$$

*if* $E_* = E_3$, *and it has two positive roots*

$$
\omega_{2,3} = \frac{-tr(J_{E_*}) \pm \sqrt{[tr(J_{E_*})]^2 - 4\det(J_{E_*})}}{2}, \qquad \tau = \tau_{2,3}^j = \frac{1}{\omega_{2,3}} \left( \frac{\pi}{2} + 2\pi j \right)
\tag{12}
$$

if $E_* = E_3$.

**Proof.** The statement follows from Equations (3), (4) and (10). Let $E_* = E_0$. Then, $\cos \omega\tau = 0$, $\sin \omega\tau = 0$, and so system (10) has no solution. Let $E_* = E_1$ or $E_* = E_2$. Then, $\cos \omega\tau = 0$, $\omega \sin \omega\tau = -tr(J_{E_*}) > 0$, yielding $\omega\tau = \pi/2$ and $\omega = -tr(J_{E_*})$. Finally, let $E_* = E_3$ and consider the cases $-\omega^2 + \det(J_{E_*}) = 0$, $\cos \omega\tau \neq 0$ and $-\omega^2 + \det(J_{E_*}) \neq 0$. $\cos \omega\tau = 0$, together with $[\omega^2 + \det(J_{E_*})] \sin \omega\tau = -tr(J_{E_*})\omega$. □

Let $\omega_* \in \{\omega_0, \omega_1, \omega_2, \omega_3\}$ be a root of system (10) and $\tau_*$ the corresponding value of $\tau$. To perform the delay Hopf bifurcation theorem, we need to guarantee simple root and transversality at $\lambda = i\omega_*$ and $\tau = \tau_*$, respectively.

**Proposition 2.** *The characteristic equation* (7) *admits a pair of simple conjugate pure imaginary roots* $\lambda = \pm i\omega_*$ *at* $\tau = \tau_*$. *The crossing direction of the pair of simple conjugate pure imaginary roots through the imaginary axis is determined by*

$$sign \left[ \frac{dRe(\lambda)}{d\tau} \right]_{\tau=\tau_*} > 0.$$

**Proof.** Differentiating (8) with respective to $\tau$ we get

$$\left\{ 2\lambda e^{\lambda\tau} - tr(J_{E_*}) - \tau \left[ \det(J_{E_*})e^{-\lambda\tau} - \lambda^2 e^{\lambda\tau} \right] \right\} \left( \frac{d\lambda}{d\tau} \right) = \lambda \left[ \det(J_{E_*})e^{-\lambda\tau} - \lambda^2 e^{\lambda\tau} \right]. \tag{13}$$

If $\lambda = i\omega_*$ were a multiple root of (8), then (13) would give $i\omega_* \left[ \det(J_{E_*})e^{-i\omega_*\tau_*} + \omega_*^2 e^{i\omega_*\tau_*} \right] = 0$, leading to $[\omega_*^2 - \det(J_{E_*})] \cos \omega_*\tau_* = 0$ and $[\omega_*^2 + \det(J_{E_*})] \sin \omega_*\tau_* = 0$. Thus, $\sin \omega_*\tau_* = 0$ and so (10) yields $-tr(J_{E_*})\omega_* = 0$, i.e., an absurd. It remains to determine the direction of motion of $\lambda$ as $\tau$ is varied. From (13), we have

$$\left( \frac{d\lambda}{d\tau} \right)^{-1} = \frac{2\lambda e^{\lambda\tau} - tr(J_{E_*})}{\lambda \left[ \det(J_{E_*})e^{-\lambda\tau} - \lambda^2 e^{\lambda\tau} \right]} - \frac{\tau}{\lambda}.$$

After some calculations, we get

$$sign \left[ \frac{dRe(\lambda)}{d\tau} \right]_{\tau=\tau_*} = sign \left[ Re \left( \frac{d\lambda}{d\tau} \right)^{-1} \right]_{\tau=\tau_*} = sign \{\Gamma\},$$

where

$$\Gamma = 2\omega_*^3 + 2\det(J_{E_*})\omega_* \cos 2\omega_*\tau_* + tr(J_{E_*}) \left[ \omega_*^2 - \det(J_{E_*}) \right] \sin \omega_*\tau_*.$$

If $\omega_* = \omega_0$ $(\omega_0^2 = \det(J_{E_*}))$, one has

$$sign \{\Gamma\} = sign \left\{ 2\omega_0^3 (1 + \cos 2\omega_0\tau_0) \right\} > 0,$$

while if $\omega_* = \omega_{1,2}$ $(\cos \omega_{1,2}\tau_{1,2} = 0, \sin \omega_{1,2}\tau_{1,2} = 1)$, it is

$$sign \{\Gamma\} = sign \left\{ -tr(J_{E_*})\omega_*^2 - 4\det(J_{E_*})\omega_* - \det(J_{E_*})tr(J_{E_*}) \right\}.$$

In the latter, we have used the fact that $\omega_*^2 + \det(J_{E_*}) = -tr(J_{E_*})\omega_*$. Noticing that $-tr(J_{E_*})\omega_*^2 - 4\det(J_{E_*})\omega_* - \det(J_{E_*})tr(J_{E_*}) = 0$ has a negative discriminant, we can conclude that the sign of $\Gamma$ is the same as the sign of $-tr(J_{E_*}) > 0$. □

As each crossing of the real part of characteristic roots at $\tau_*$ is from left to right as $\tau$ increases, based on the above analysis, we have the following result.

**Theorem 1.** *Let $\omega_* \in \{\omega_1, \omega_2, \omega_3\}$ and $\tau_*^j$ ($j = 1, 2, 3$) its corresponding value of $\tau$ be defined as in Equations (11) and (12).*

(1)　　*Let $E_* = E_0, E_1$ or $E_2$. System (6) is unstable for $\tau \geq 0$.*

(2)　　*Let $E_* = E_3$. If system (10) has one positive root $\omega_* = \omega_1$ at the values $\tau_1^j$, then the equilibrium $E_*$ of system (6) is locally asymptotically stable for $\tau \in [0, \tau_1)$ and unstable for $\tau > \tau_1$. A Hopf bifurcation occurs at the equilibrium $E_*$ for $\tau = \tau_1$.*

(3)　　*Let $E_* = E_3$. If system (10) has two positive solutions $\omega_* = \omega_{2,3}$, $\omega_2 < \omega_3$, then the equilibrium $E_*$ of system (6) is locally asymptotically stable for $\tau \in [0, \hat{\tau})$ and unstable for $\tau > \hat{\tau}$, where $\tilde{\tau} = \min \left\{ \tau_{2,3}^j, j = 0, 1, 2, ... \right\}$. System (6) undergoes a Hopf bifurcation at the equilibrium $E_*$ for $\tau = \tilde{\tau}$.*

*3.2. Existence of Equilibria and Local Bifurcations with Heterogeneous Time Delays*

The aim is to extend the analysis developed in the previous section when $\tau_1 \neq \tau_2$, $\tau_1 \geq 0$ and $\tau_2 \geq 0$, in system (5), and the equilibrium point $E_*$ is the Nash equilibrium $E_3$. The Jacobian matrix evaluated at $E_3$ leads us to the following characteristic equation,

$$
\begin{vmatrix} -\lambda + \left[ \left( \dfrac{4Bb}{9} - \gamma \right) \alpha_1 x_1^* \right] e^{-\lambda \tau_1} & \left[ \dfrac{4Bb\alpha_1 x_1^*}{9} \right] e^{-\lambda \tau_1} \\[4mm] \left[ \dfrac{4Bb\alpha_2 x_2^*}{9} \right] e^{-\lambda \tau_2} & -\lambda + \left[ \left( \dfrac{4Bb}{9} - \gamma \right) \alpha_2 x_2^* \right] e^{-\lambda \tau_2} \end{vmatrix} = 0,
$$

namely,

$$
\lambda^2 + M\alpha_1 \lambda e^{-\lambda \tau_1} + M\alpha_2 \lambda e^{-\lambda \tau_2} + \det(J_{E_3}) e^{-\lambda(\tau_1 + \tau_2)} = 0, \tag{14}
$$

where

$$
\det(J_{E_3}) = \frac{16A^2 \gamma \alpha_1 \alpha_2}{9(9\gamma - 8Bb)} > 0, \qquad M = \frac{4A(9\gamma - 4Bb)}{9(9\gamma - 8Bb)} > 0.
$$

3.2.1. Case $\tau_1 = 0, \tau_2 > 0$

Equation (14) reduces to

$$
\lambda^2 + M\alpha_1 \lambda + \left[ M\alpha_2 \lambda + \det(J_{E_3}) \right] e^{-\lambda \tau_2} = 0. \tag{15}
$$

In absence of delay, i.e., $\tau_2 = 0$, $E_3$ is stable. With the time delay $\tau_2$ varying, system (5) will lose the stability. To obtain such critical values of time delay, supposing $\lambda = i\omega$, $\omega > 0$, is a purely imaginary root of (14), one has

$$
\omega^2 = M\alpha_2 \omega \sin \omega \tau_2 + \det(J_{E_3}) \cos \omega \tau_2, \qquad -M\alpha_1 \omega = -\det(J_{E_3}) \sin \omega \tau_2 + M\alpha_2 \omega \cos \omega \tau_2. \tag{16}
$$

Taking the square, adding the equations and performing some simplification processes, and setting $z = \omega^2$, we have

$$
z^2 + M^2 \left( \alpha_1^2 - \alpha_2^2 \right) z - \left[ \det(J_{E_3}) \right]^2 = 0. \tag{17}
$$

Obviously, if (17) has no positive solution for $z$, then (15) cannot have purely imaginary roots. Noticing that $-\left[ \det(J_{E_3}) \right]^2 < 0$, it follows that Equation (17) has a unique positive root $\omega_+$, where

$$
\omega_+ = \sqrt{\frac{-M^2 \left( \alpha_1^2 - \alpha_2^2 \right) + \sqrt{M^4 \left( \alpha_1^2 - \alpha_2^2 \right)^2 + 4 \left[ \det(J_{E_3}) \right]^2}}{2}}.
$$

Solving (16) for $\sin \omega \tau_2$ and $\cos \omega \tau_2$, we get

$$\sin \omega \tau_2 = \frac{\left[\alpha_1 \det(J_{E_3}) + \alpha_2 \omega^2\right] M \omega}{\left[\det(J_{E_3})\right]^2 + M^2 \alpha_2^2 \omega^2}, \qquad \cos \omega \tau_2 = \frac{\left[\det(J_{E_3}) - M^2 \alpha_1 \alpha_2\right] \omega^2}{\left[\det(J_{E_3})\right]^2 + M^2 \alpha_2^2 \omega^2}.$$

As $sign\left(\cos \omega \tau_2\right) = sign(\det(J_{E_3}) - M^2 \alpha_1 \alpha_2) = sign(-72\gamma^2 - 16B^2 b^2)$, one has the following sequence of critical delays

$$\tau_{2,j}^+ = \frac{1}{\omega_+} \left\{ 2\pi - \sin^{-1} \left\{ \frac{\left[\alpha_1 \det(J_{E_3}) + \alpha_2 \omega^2\right] M \omega}{\left[\det(J_{E_3})\right]^2 + M^2 \alpha_2^2 \omega^2} \right\} + 2j\pi \right\}, \tag{18}$$

where $j = 0, 1, 2, \ldots$

**Lemma 2.** *When $\tau = \tau_{2,j}^+$, then (15) has a pair of pure imaginary roots $\pm i\omega_+$.*

We next detect the stability switch at which the equilibrium loses stability. As $\lambda$ is a function of delay $\tau_2$, we need the minimum solution of for which a derivative of $\lambda(\tau_2)$ is positive. By selecting $\tau_2$ as the bifurcation parameter and differentiating the characteristic equation (15), with respect, we get

$$\left\{ 2\lambda + M\alpha_1 + M\alpha_2 e^{-\lambda \tau_2} + \left[-M\alpha_2 \lambda - \det(J_{E_3})\right] \tau_2 e^{-\lambda \tau_2} \right\} \left(\frac{d\lambda}{d\tau_2}\right) = \left[M\alpha_2 \lambda + \det(J_{E_3})\right] \lambda e^{-\lambda \tau_2}. \tag{19}$$

We now prove $\lambda = i\omega_+$ to be a simple root for (15). If this root is repeated, then (19) implies $\left[M\alpha_2 i\omega_+ + \det(J_{E_3})\right] i\omega_+ e^{-i\omega_+ \tau_{2,j}^+} = 0$, i.e., a contradiction. Next, we can obtain that

$$\left(\frac{d\xi}{d\tau_2}\right)^{-1} = \frac{(2\lambda + M\alpha_1) e^{\lambda \tau_2} + M\alpha_2}{\left[M\alpha_2 \lambda + \det(J_{E_3})\right] \lambda} - \frac{\tau_2}{\lambda}.$$

Therefore, we have

$$sign \left\{ \frac{d\left(Re\lambda\right)}{d\tau_2} \bigg|_{\tau = \tau_{2,j}^+} \right\} = sign \left\{ Re \left(\frac{d\lambda}{d\tau_2}\right)^{-1}_{\tau = \tau_{2,j}^+} \right\}$$

$$= sign \left\{ \frac{M^2 \alpha_2^2 \omega_+^4 + 2\left[\det(J_{E_3})\right]^2 \omega_+^2 + M^2 \alpha_1^2 \left[\det(J_{E_3})\right]^2}{\left\{ M^2 \alpha_2^2 \omega_+^2 + \left[\det(J_{E_3})\right]^2 \right\} \left(\omega_+^2 + M^2 \alpha_1^2\right) \omega_+^2} \right\} > 0.$$

This inequality implies that the real parts of complex eigenvalues of (15) turn to be positive from negative when crosses the imaginary axis as $\tau_2$ increases. The previous analysis can be summarized as follows.

**Theorem 2.** *Let $\tau_{2,j}^+$ $(j = 0, 1, 2, \ldots)$ be defined as in (18). The equilibrium $E_3$ of system (5) is locally asymptotically stable for $\tau_2 \in [0, \tau_{2,0}^+)$, unstable for $\tau > \tau_{2,0}^+$ and undergoes a Hopf bifurcation at the equilibrium at $\tau_2 = \tau_{2,0}^+$.*

### 3.2.2. Case $\tau_1 > 0$, $\tau_2$ Fixed in Its StableInterval

To investigate the effect of multiple delays on the local stability of equilibrium point $E_3$, we regard $\tau_1$ as the varying parameter for any fixed delay $\tau_2$ in its stable interval, i.e. $\tau_2 \in [0, \tau_{2,0}^+)$. Let $\lambda = i\omega$ ($\omega > 0$) be a root of the characteristic Equation (14), then

$$- \omega^2 + M\alpha_1 i\omega(\cos \omega\tau_1 - i\sin \omega\tau_1) + M\alpha_2 i\omega(\cos \omega\tau_2 - i\sin \omega\tau_2)$$
$$+ \det(J_{E_3}) \left[\cos \omega (\tau_1 + \tau_2) - i\sin \omega (\tau_1 + \tau_2)\right] = 0.$$

Therefore, it follows

$$\omega^2 - M\alpha_1\omega \sin \omega\tau_1 - M\alpha_2\omega \sin \omega\tau_2 \;=\; \det(J_{E_3}) \cos \omega (\tau_1 + \tau_2), \tag{20}$$

$$-M\alpha_1\omega \cos \omega\tau_1 - M\alpha_2\omega \cos \omega\tau_2 \;=\; -\det(J_{E_3}) \sin \omega (\tau_1 + \tau_2). \tag{21}$$

Squaring and adding these equations yields

$$\left(\omega^2 - M\alpha_1\omega \sin \omega\tau_1 - M\alpha_2\omega \sin \omega\tau_2\right)^2 + (M\alpha_1\omega \cos \omega\tau_1 + M\alpha_2\omega \cos \omega\tau_2)^2 = \left[\det(J_{E_3})\right]^2,$$

so that we get

$$g(\omega) = \omega^4 + (-2M\alpha_1 \sin \omega\tau_1 - 2M\alpha_2 \sin \omega\tau_2)\omega^3$$
$$+ \left[M^2\alpha_1^2 + M^2\alpha_2^2 + 2M^2\alpha_1\alpha_2 \cos \omega (\tau_1 - \tau_2)\right]\omega^2 - \left[\det(J_{E_3})\right]^2 = 0. \tag{22}$$

It is easy to see that $g(0) = -\left[\det(J_{E_3})\right]^2 < 0$ and $g(\omega) = +\infty$ as $t \to +\infty$. Thus, $g(\omega) = 0$ has at least one positive solution. Assume Equation (22) has finitely many positive solutions and denote them by $\omega_1, \omega_2, ..., \omega_N$. For every fixed $\omega_l$, $l = 1, 2, ..., N$, we can derive from (20) and (21) the sequence of critical values $\tau_{1,l}^j > 0$ ($j = 1, 2, ...$). Let

$$\tilde{\tau}_1 = \min \left\{\tau_{1,l}^j, l = 1, 2, ..., N, j = 0, 1, 2, ...\right\}. \tag{23}$$

For $\tau_1 = \tilde{\tau}_1$, Equation (14) has a pair of purely imaginary roots $\lambda = \pm i\tilde{\omega}$. Let $\lambda(\tau_1)$ be the root of (14) near $\tau_1 = \tilde{\tau}_1$ such that $Re(\lambda(\tilde{\tau}_1)) = 0$ and $Im(\lambda(\tilde{\tau}_1)) = \tilde{\omega}$. Then, taking differentiation of both sides of (14) with respect to $\tau_1$, we have

$$\left[2\lambda + M\alpha_1 e^{-\lambda\tau_1} + M\alpha_2 e^{-\lambda\tau_2} - M\alpha_2\lambda\tau_2 e^{-\lambda\tau_2} - \det(J_{E_3})(\tau_1 + \tau_2)e^{-\lambda(\tau_1+\tau_2)} - M\alpha_1\tau_1\lambda e^{-\lambda\tau_1}\right]\left(\frac{d\lambda}{d\tau_1}\right)$$
$$= \lambda \left[M\alpha_1\lambda e^{-\lambda\tau_1} + \det(J_{E_3})e^{-\lambda(\tau_1+\tau_2)}\right]. \tag{24}$$

Now, notice that from (24) it follows that the root $\lambda = \pm i\tilde{\omega}$ is simple. Otherwise,

$$i\tilde{\omega} \left[M\alpha_1 i\tilde{\omega}e^{-i\tilde{\omega}\tilde{\tau}_1} + \det(J_{E_3})e^{-i\tilde{\omega}(\tilde{\tau}_1+\tau_2)}\right] = 0$$

and (14) evaluated at $\lambda = i\tilde{\omega}$ give $i\tilde{\omega}\left(i\tilde{\omega} + M\alpha_2 e^{-i\tilde{\omega}\tau_2}\right) = 0$, namely, $i(\tilde{\omega} - M\alpha_2 \sin \tilde{\omega}\tau_2) + M\alpha_2 \cos \tilde{\omega}\tau_2 = 0$. This means we have $\tilde{\omega} = M\alpha_2 \sin \tilde{\omega}\tau_2$ and $\cos \tilde{\omega}\tau_2 = 0$, i.e. $\tilde{\omega} = M\alpha_2$, $\sin \tilde{\omega}\tau_2 = 1$ and $\cos \tilde{\omega}\tau_2 = 0$. On the other hand, using the formulas $\cos \omega (\tilde{\tau}_1 + \tau_2) = \cos \omega\tilde{\tau}_1 \cos \omega\tau_2 -$

$\sin \omega \tilde{\tau}_1 \sin \omega \tau_2$ and $\sin \omega (\tilde{\tau}_1 + \tau_2) = \sin \omega \tilde{\tau}_1 \cos \omega \tau_2 + \cos \omega \tilde{\tau}_1 \sin \omega \tau_2$ in (20),(21) leads to $\sin \tilde{\omega} \tilde{\tau}_1 = 0$ and $\cos \tilde{\omega} \tilde{\tau}_1 = 0$, which is a contradiction. Then, from (24), we find

$$\left( \frac{d\lambda}{d\tau_1} \right)^{-1} = \frac{2\lambda + M\alpha_1 e^{-\lambda\tau_1} + M\alpha_2 e^{-\lambda\tau_2} - M\alpha_2 \lambda \tau_2 e^{-\lambda\tau_2} - \det(J_{E_3})\tau_2 e^{-\lambda(\tau_1+\tau_2)}}{\lambda \left[ M\alpha_1 \lambda e^{-\lambda\tau_1} + \det(J_{E_3}) e^{-\lambda(\tau_1+\tau_2)} \right]} - \frac{\tau_1}{\lambda}, \quad (25)$$

which can be rewritten by (15) as

$$\left( \frac{d\lambda}{d\tau_1} \right)^{-1} = -\frac{2\lambda + M\alpha_1 e^{-\lambda\tau_1} + M\alpha_2 e^{-\lambda\tau_2} + \tau_2 \left( \lambda^2 + M\alpha_1 \lambda e^{-\lambda\tau_1} \right)}{\lambda \left( \lambda^2 + M\alpha_2 \lambda e^{-\lambda\tau_2} \right)} - \frac{\tau_1}{\lambda}.$$

Consequently, Equation (25) becomes

$$\left( \frac{d\lambda}{d\tau_1} \right)^{-1}_{\tau_1 = \tilde{\tau}_1} = \frac{P + iR}{\tilde{\omega} (Q - iS)} - \frac{\tilde{\tau}_1}{i\tilde{\omega}},$$

where

$$P = M\alpha_1 \cos \tilde{\omega} \tilde{\tau}_1 + M\alpha_2 \cos \tilde{\omega} \tau_2 - \tau_2 \left( -M\alpha_1 \tilde{\omega} \sin \tilde{\omega} \tilde{\tau}_1 + \tilde{\omega}^2 \right), \qquad Q = M\alpha_2 \tilde{\omega} \cos \tilde{\omega} \tau_2$$

$$R = 2\tilde{\omega} - M\alpha_1 \sin \tilde{\omega} \tilde{\tau}_1 - M\alpha_2 \sin \tilde{\omega} \tau_2 + M\alpha_1 \tilde{\omega} \tau_2 \cos \tilde{\omega} \tilde{\tau}_1, \qquad S = M\alpha_2 \tilde{\omega} \sin \tilde{\omega} \tau_2 - \tilde{\omega}^2.$$

Therefore, we have

$$sign \left[ \frac{dRe(\lambda)}{d\tau_1} \right]_{\tau_1 = \tilde{\tau}_1} = sign \left[ Re \left( \frac{d\lambda}{d\tau_1} \right)^{-1} \right]_{\tau_1 = \tilde{\tau}_1} = sign \{ G(\tilde{\omega}, \tilde{\tau}_1) \},$$

where

$$G(\tilde{\omega}, \tilde{\tau}_1) = PQ - RS = \left( 2\tilde{\omega}^2 + M^2 \alpha_2^2 \right) \tilde{\omega} - M^2 \alpha_1^2 \tilde{\omega} \sin \tilde{\omega} \tilde{\tau}_1$$

$$- 3M\alpha_2 \tilde{\omega}^2 \sin \tilde{\omega} \tau_2 + M^2 \alpha_1 \alpha_2 \tilde{\omega} \cos \tilde{\omega}(\tilde{\tau}_1 - \tau_2)$$

$$+ \tau_2 \left\{ M\alpha_1 \tilde{\omega}^3 \cos \tilde{\omega} \tilde{\tau}_1 - M\alpha_2 \tilde{\omega}^3 \cos \tilde{\omega} \tau_2 + M^2 \alpha_1 \alpha_2 \sin \tilde{\omega}(\tilde{\tau}_1 - \tau_2) \right\}. \quad (26)$$

The pair of purely imaginary roots crosses the imaginary axis from left (resp. right) to right (resp. left) at $\tilde{\tau}_1$ if the sign of $PQ - RS$ is positive (resp. negative). Based on the found transversality condition and the Hopf bifurcation theorem, one has the following assertions.

**Theorem 3.** *Let $\tilde{\tau}_1$ and $G(\tilde{\omega}, \tilde{\tau}_1)$ be defined as in (23) and (26).*

(1) *If Equation (22) exhibits one single positive root satisfying $sign \{ G(\tilde{\omega}, \tilde{\tau}_1) \} \neq 0$, then the equilibrium $E_3$ of system (5) is locally asymptotically stable for $\tau_1 \in [0, \tilde{\tau}_1)$ and system (5) displays a Hopf bifurcation from $E_3$ if $\tau_1 = \tilde{\tau}_1$ when $sign \{ G(\tilde{\omega}, \tilde{\tau}_1) \} > 0$,; it is locally asymptotically stable for $\tau_1 \geq 0$ if $sign \{ G(\tilde{\omega}, \tilde{\tau}_1) \} < 0$.*

(2) *If Equation (22) presents at least two positive roots, then there exists some delayed interval sequence where the equilibrium $E_3$ of system (5) is locally asymptotically stable. The dynamical behaviour of system (5) near $E_3$ switches from stability to instability, and back again as time delays increase beyond the critical values, and Hopf bifurcations may occur.*

## 4. Conclusions

This paper extends the discrete-time two-stage game of R&D competition between two high-tech enterprises of Zhou and Wang [29] to the case of continuous-time version with delays. The use of delay

differential equations makes it possible to go beyond some limitations of other modelling approaches in a natural way. It is found that the boundary equilibrium points are always unstable and the Nash equilibrium loses its stability. The emergence of Hopf bifurcations is also characterised. Our findings, therefore, stress how the extent of time delays may be responsible for the existence of interesting dynamic outcomes, and underline the importance of the theoretical modelling framework used as a tool that may dramatically change the long-term findings of an economy.

**Author Contributions:** All the authors have equal contribution to this study. All authors have read and agreed to the published version of the manuscript.

**Funding:** This research received no external funding.

**Conflicts of Interest:** The authors declare no conflict of interest. The funders had no role in the design of the study; in the collection, analyses, or interpretation of data; in the writing of the manuscript, or in the decision to publish the results.

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
