# Peer review of "Bifurcation Analysis of a Duopoly Game with R&D Spillover, Price Competition and Time Delays"

_symmetry, doi:10.3390/sym12020257_

Round 1

Reviewer 1 Report

The topic is interesting, the paper is well-written and correct. I have just two minor remarks to improve it.

1) When the Authors introduce the problem, on page 2 they present the original discrete model on which they build the continuous version. The final discrete model is just reported (eq. 1). I think that the paper would benefit from a wider introduction of such model, spending a few words about the discrete duopoly game model of R&D competition that gives rise to the system 2) The paper would benefit from a numerical section in which present what may happen if the equilibrium loses its stability, briefly investigating the developement of the bifurcation. If this is possible, I encourage the Authors to add a Section

Author Response

We would to thanks the Referee for their useful suggestions. The first one was completely satisfied with an addition to the introduction. For that concerns the second request about the possibility in adding some numerical examples, in our opinion this option is out of the scope the theoretical mission of the study.

Reviewer 2 Report

I wish you gave some examples. In particular, I would like to see how the continuous-time results differ from the discrete-time results.

Author Response

The examples are tautological in our opinion. The algebraic transformation from the discrete form to the continuous-type is so clear

Reviewer 3 Report

Overall I think this paper makes an interesting contribution to the literature.

My main comment would be the need to motivate the modelling of delay and discuss the interpretation of the results. At the moment it reads very much like a technical exercise and yet this literature is addressing a very practical problem. So, what is the interpretation of the delay? And what is the interpretation of chaotic dynamics?

A more minor point - Proposition 1 could do with rewriting because it is very hard to follow which solution corresponds to which equilibrium. To split into parts (1), (2) and (3) would help.  

Author Response

We thanks the Referee for their so useful comments. All suggestions were deeply considered and inserted on the revised version.